# The Sizes of Spine Curvatures of Children That Practice Selected Sports

**DOI:** 10.3390/ijerph20031826

**Published:** 2023-01-19

**Authors:** Natalia Twarowska-Grybalow, Aleksandra Truszczyńska-Baszak

**Affiliations:** Department of Physiotherapy, Józef Piłsudski University of Physical Education in Warsaw, 00-968 Warsaw, Poland

**Keywords:** body posture, Moiré method, physical activity, spine

## Abstract

(1) Background: The aim of this study was to evaluate the shape of the spine curves in the cervical, thoracic and lumbar sections of children that practice selected sports. (2) Methods: The body posture of the examined children was assessed using the digital photography method, i.e., the Moiré method. Selected parameters characterizing the curvature of the spine (the Alpha, Beta and Gamma angles, the size of kyphosis in the thoracic spine and the size of lordosis in the lumbar spine) were analyzed. (3) Results: The study of the body posture using the Moiré method allowed for the assessment of the angles that determine the size of the spine’s curvature. The analysis of differences among the groups included in the study (football, swimming, biathlon/taekwondo, volleyball) was carried out on the basis of one-dimensional models that take into account the distributions of individual parameters. On the basis of the Alpha, Beta and Gamma angles, it was possible to calculate the size of kyphosis in the thoracic section and the size of lordosis in the lumbar spine. There was a statistically significant difference in the size of the Alpha, Beta and Gamma parameters among the groups. (4) Conclusions: Most of the respondents had the correct body posture in the sagittal plane, regardless of the type of sport they practiced. Our results did not allow us to unequivocally state whether practicing various sports and having different training loads resulting from these sports have a negative or positive effect on the size of the anterior–posterior curvatures of the spine.

## 1. Introduction

The correct body posture is characterized by slight curvatures of the spine which are shaped by the cooperation of muscles antagonistic to the gravitational force [1,2]. Until the age of 7, the curves of the spine do not take their final shape. During this period, a slight dominance of kyphosis in the thoracic spine can be noticed. Around the age of 7, correct lordosis begins to form in the cervical spine, and between the ages of 8 and 11, normal lordosis in the lumbar region is produced [3].

The norms that determine the correct shape of the spine for children are established on the basis of their age. The deviations from those norms may mean deepening, flattening or asymmetry of curves in different planes [2,4].

A review of the current literature does not indicate clearly whether competitive practice in various sports has an impact on the size of the anterior–posterior curvatures of the spine. Each sport has different and specific characteristics. The differences are noticed in the training plans and in the impact of the training on the athlete’s body. Solving the problem of how training and various training loads affect the shape of the spine of young players is important because over years of practicing sports, changes in the body become permanent and affect the next stages of ontogenesis [5]. Increasing deviations within the spine may lead in the future to back pain and may increase the risk of injuries and overload the musculoskeletal system [6]. The literature indicates both the positive effect of increased physical activity on health [7,8,9] as well as the risk of posture asymmetry and pain in the lower back of the spine of children who practice sports such as volleyball, athletics, team sports, martial arts and gymnastics [10,11,12,13].

The early-school period of the growth jump is characterized by a child’s weakened muscle strength, poor stabilization and lability of the physiological curvatures of the spine [3]. The instability of the spine’s curvatures is related to the fact that in the case of abnormal muscle tone or external factors, such as changes in lifestyle related to initiation of learning at school, they may be incorrectly shaped. The plasticity of changes during this period may also be beneficial. By using properly implemented prophylaxis and corrective exercises, posture defects can be corrected. Uncorrected spine curvature in the above-mentioned period may lead to the progression of negative changes in the body posture during puberty [14,15,16].

Currently, a commonly used imaging test for evaluating deviations within the spine, which directly affects the whole-body posture, is a radiographic examination. This is the gold standard method for the analysis of the skeletal system [17,18]. X-ray examination is accurate and reliable, but is associated with radiation and negative health effects resulting from frequent exposure. The projection method with the Moiré system is a method of assessing body posture whose reliability has been confirmed by numerous publications in recent years [6,17,18,19]. This instrumental analysis uses an optical phenomenon, that is the process of refraction of light beams, which enables the creation of a three-dimensional image from a two-dimensional image [17]. Other instrumental investigations, such as 3D-motion analysis, quantitatively and three-dimensionally evaluate sagittal posture and spine in upright standing and during walking [20,21]. The digital photography Moiré method and 3D-Motion analysis do not use radiation and show a good accuracy; the former has shorter test execution times, and the latter also allows for a dynamic study. However, they are supplementary to radiological examinations.

The aim of this study was to evaluate the shape of the spine curves in the cervical, thoracic and lumbar sections of children that practice selected sports.

## 2. Material and Methods

The study group consisted of 247 training children who attended sport-profile public primary schools or who train in a selected discipline in a sports club. The control group consisted of 63 non-training children who attended public primary school. The selected sports were:(a)Football (60 persons);(b)Swimming (64 persons);(c)Biathlon/taekwondo (62 persons attending the same class in a primary school with a sports profile. All people participated in these physical education classes and had the same general development training. Only the training of the leading sport discipline differed, and the number of training units depended on the age of the trainees);(d)Volleyball (61 persons).

The age of the examined children varied from 8 to 13 years. The detailed data of the number of participants and their age, weight and height, as well as their BMI index, are presented in Table 1.

The inclusion criteria for the study were:Obtaining the written consent of the legal guardian to conduct the examination;Attending a sport-profile public primary school or training in a given sport in a club for 6 h a week as a minimum (study group);The age of the child being in the range 8.5–13.5 years.The exclusion criteria for the study were:Severe orthopedic injuries (fracture of the lower limb, inability to fully load the limb);Asymmetry of the length of the lower limbs above 2 cm (assessment by physical examination);Structural scoliosis (assessment by physical examination);The presence of other diseases causing musculoskeletal deformities or structural postural defects;Failure to obtain the written consent of the legal guardian to conduct the examination.

The body posture of the examined children was assessed using the digital photography Moiré method. The guidelines for carrying out the body posture test with the Moiré 4G system and for entering data into the CQ-Posture program are consistent with the guidelines of the device’s manufacturer.

Selected parameters characterizing the curvature of the spine (Alpha, Beta and Gamma angles; KPP; KLL) were analyzed. The above parameters are marked in Figure 1 and are characterized by:Alpha angle—the inclination of the lumbosacral spine;Beta angle—the inclination of the thoracolumbar segment;Gamma angle—the inclination of the upper thoracic spine;KPP—the size of kyphosis in the thoracic spine;KLL—the size of lordosis in the lumbar spine;

A survey was used to analyze the training plan of each sport, the number of training units carried out in the presence of the trainer and carried out alone, and the declared willingness to train in the selected sport in the future. The questionnaire also contained information about the amount of time the child spent additionally on individual or organized recreational out-of-school physical activity, and how much time per day a child spent passively in front of electronic devices.

All measurements were preceded by obtaining written consent from the parents or legal guardians, and after obtaining the consent of the director of the facility or of the sports club coach. The tests were carried out in the morning during school activities or at the time that was pre-determined with the “Sparta” and “Escola Varsovia” sports clubs. The research took place from March 2017 to March 2020. The methodology of this study is entirely based on the pediatric measurement protocol and standards that are in line with the Helsinki Declaration for human research. Participation in the study was voluntary and in the event of the child’s refusal, the study was not conducted. Figure 2 presents a diagram showing the participation of respondents in the subsequent stages of the project.

Statistical significance was calculated on the basis of the actual size of the effects. Sizes of effects were defined as differences between the mean values of the analyzed variables. The repeatability of the measurements of the parameters that determine the body posture was based on the intragroup correlation coefficient ICC (Intraclass Correlation Coefficient). The *p* = 0.05 was taken as the borderline significance level.

## 3. Results

The study of the body posture using the Moiré method allowed for the assessment of the size of the angles that determine the size of the spine’s curvature. The analysis of differences between the studied groups was carried out on the basis of one-dimensional models that take into account the distributions of individual parameters. On the basis of the angles Alpha, Beta, Gamma, it was possible to calculate the size of kyphosis in the thoracic section and the size of lordosis in the lumbar spine. There was a statistically significant difference in the size of the Alpha, Beta and Gamma parameters among the groups. The detailed data concerning the Alpha, Beta and Gamma angles and the size of the spine curvatures are presented in Table 2.

Based on the mean values of angles and the size of curvatures in the spine, the types of body posture and the presence of correct or incorrect body posture in the study groups were determined using the modified Wolański method by Zeyland-Malawka [22]. A kyphotic posture dominated in the group that trained biathlon/taekwondo, football and swimming. This was present in 24 (38.7%) children in the biathlon/taekwondo group, in 29 (48.3%) children in the football training group and in 29 (46.0%) in the swimming group. In the group that trained volleyball, an equivalent posture was dominant and it was found for 27 (44.3%) children. In the control group, the kyphotic posture and the equivalent posture were found in the same number of children and occurred in 23 participants (36.5%). The detailed data are presented in Figure 3. The distributions of the types of attitudes in the studied groups did not differ significantly (*p* = 0.1779).

The kyphotic, lordotic and equivalent posture was also analyzed based on the modified Wolański method by Zeyland-Malawka and the attitudes were divided into correct and defective based on the exact types of posture (I, II, III). The correct body posture is determined by type I of the equivalent posture, type II of the kyphotic posture and type I of the lordotic posture. Abnormal body posture was found only in four participants (6.6%) practicing volleyball, three (4.7%) practicing swimming and for two (3.2%) in the control group. For the remaining participants, the correct proportion of the individual spine curves was found. The detailed data are presented in Figure 4.

### Analysis of Multivariate Models

After a more detailed analysis of the Alpha parameter in the multivariate model, a relationship between the Alpha angle and extracurricular activity in the recreational form (*p* = 0.0320) and sex (*p* = 0.0044) was demonstrated. The value of Alpha angle in the model grew with the increase in the number of extracurricular activities per week in all groups. A relationship between the value of the Alpha angle and sex in individual groups was also noticed. In the group that trained swimming and biathlon/taekwondo and also in the control group, higher values of the Alpha angle were found for girls. The exception was the group training volleyball, where the higher average value of the Alpha angle was found for boys. There was no correlation between the value of the Alpha angle and somatic parameters, age and other tested parameters related to lifestyle and physical activity.

After a more detailed analysis of the Beta parameter in the multivariate model, a relationship was demonstrated between the Beta angle and the training load of selected sports per week (*p* = 0.0083), the number of hours per week of recreational extracurricular activity (*p* = 0.0129), sex (*p* = 0.0097) and BMI values (*p* = 0.0154). The Beta angle decreased with increasing training load. The exception was the swimming group, in which the Beta angle slightly increased with more training hours. The changes in values were not clinically significant. A relationship between the Beta angle and out-of-school activity was observed in the multivariate model. As the number of hours of recreational physical activity per week increased, the Beta angle decreased for the examined children. The changes in values were not clinically significant. The Beta angle value obtained in the multivariate model was higher for girls than for boys in the volleyball training group, swimming and the control group. On the other hand, in the biathlon/taekwondo training group, the Beta angle was lower for boys. The Beta angle showed in the multivariate model higher values for overweight children. The exception was the swimming group. The differences were not clinically significant.

The multivariate analysis also showed a relationship between the Gamma angle and the sex of the examined persons (*p* = 0.0145). Higher values of the Gamma angle were found for boys in all groups except for the biathlon/taekwondo training group.

After a more detailed analysis of the size of kyphosis in the thoracic section in the multivariate model, a relationship between the size of kyphosis and the training load of selected sports per week (*p* = 0.0289) and BMI values (*p* = 0.0244) was demonstrated. The size of kyphosis decreased with increasing training load. The exception was the swimming group, in which the Beta angle slightly increased with more training hours. The changes in values were not clinically significant. The size of kyphosis in the multivariate model was lower for overweight children. The exception was the football and swimming training groups. The differences were not clinically significant.

The statistical model also showed a relationship between the magnitude of lordosis in the lumbar region and sex (*p* = 0.0003). The size of lordosis was greater for boys in all groups studied. The obtained differences were not clinically significant.

## 4. Discussion

The analysis of the current state of knowledge did not give an unequivocal answer to the question of what influence physical activity and practicing various sports in school-age children have on the size of spine curvatures. Many publications have confirmed the positive effect of sports on body posture and the shaping of spine curves [7,8]. On the other hand, researchers have reported negative consequences of asymmetric physical activity for various elements of the body posture [5,11,13,23,24,25,26,27,28,29]. There is also scientific evidence that indicates in the last decade a worldwide increase in the percentage of posture defects and abnormal spine curvatures in children and adolescents worldwide, regardless of physical activity [14,15,16,30,31,32].

Understanding the relationship between physical activity and the shape of the spine curves may be beneficial in the process of training and appropriate prophylaxis in maintaining the correct body posture, especially when posture defects may develop within a few weeks [16]. If diagnosed early, they can be corrected, and if left untreated, they lead to permanent deformities and can negatively affect the training process and sports performance of a child.

The normative ranges for the anterior–posterior spine shape are an attempt to assess the correct values for a given population. In order to establish the normative values for the Alpha, Beta and Gamma angles, Łubkowska [33] examined 1223 children, including 609 girls and 614 boys aged 7–15 years old. The results of their research allowed the determination of the average values of the Alpha, Beta and Gamma angles, i.e., the angles defining the curvature of the spine. The values of the angles were within the normative ranges given for a specific age. The mean value of the Beta angle for the examined children in our own research was lower than the given normative ranges. The lower value of the Beta angle indicates the flattening of kyphosis in the thoracic spine, which was characteristic of the groups in our own research. The differences between the individual groups were not clinically significant. On the other hand, Grabara et al. [34] studied 331 girls and 286 untrained boys aged 8–16 years old. The size of kyphosis in the thoracic section (sum of Beta and Gamma angles) and lordosis in the lumbar section (sum of Alpha and Beta angles) were also checked. Comparing the results of the study with the results of our own research, it can be stated that the angles of the spine curvature in this study were lower than the values of the spine curvature given by Grabara. The differences were especially noticeable in the case of the Beta angle and the size of kyphosis. In our research, the examined children had reduced kyphosis in the thoracic section. However, this was not associated with practicing any sports and with increased physical activity in their free time.

The authors also analyzed the size of anterior–posterior curvatures for people training in various sports. Grabara [25] compared the Alpha, Beta and Gamma angles for young people training volleyball versus a control group aged 14–16 years. The group of 104 athletes and 114 untrained boys was divided into three subgroups depending on the age of the study participants. The 14-year-old boy subgroup consisted of 28 athletes and 44 non-training participants. The values obtained in the above study, in comparison with the results of our own research, show greater kyphosis in the thoracic segment for both training and non-training adolescents in the Grabara study. In both studies, the Alpha and Gamma angles were greater for those who did not practice any sports discipline. In our own research, children training volleyball were characterized by a larger Alpha angle, and a much smaller Beta and Gamma angle. The differences in the size of the spine curvatures in our own research and in Grabara’s work may result from the age and sex of the respondents. In Grabara’s work, study participants were between 14 and 16 years of age. In this research, the average age of children training volleyball was 11.76 years, and the average age of non-training children was 11.25 years. In the Grabara study, only the male participants took part. In the authors’ own research, there were 44 girls (72.1%) and 17 boys (27.9%) training in volleyball. As shown by other studies [25,33], there are significant differences in the normative ranges of the anterior–posterior curvatures in both sexes and at different ages.

Another criterion that could explain the differences between our results and that of Grabara [25] was the training period. In our results, children had been training volleyball for about 2 years, and in Grabara’s work, the criterion for inclusion in the study was a training period varying from 2 to 6 years. However, it cannot be clearly stated how the differences between Grabara’s results and our results were influenced by different numbers of volleyball training sessions.

An analysis of the body posture of players training football was also made by Grabara [27]. She enrolled 73 children aged 11–14 that trained in football and 78 non-training persons of the same age. The training period was 2 years for those aged 11 and 4 years for those aged 13. The number of training sessions per week was 3 times for the younger age groups and 5 times for older players. The method of the study also included the study of body posture with use of the Moiré phenomenon. Statistically significant differences in the size of the lordosis angle for those training football compared to the control group were found. For training children of all ages, a reduction in lordosis in the lumbar region was observed. In our own research, no statistically significant differences were observed in the amount of lordosis in the lumbar region between the training groups and the control group. The football training group did not differ significantly from other training groups.

In another study, Grabara [25] compared the body posture of people training volleyball and non-training people using the Moiré method. She qualified for the studies 104 players aged 14–16 and their 114 non-training peers. The training period ranged from 2 to 6 years. In the sagittal plane, volleyball practitioners showed more frequent flattening of the lumbar lordosis and increased thoracic kyphosis than their non-training peers.

The limitation in this study was the inability to test the group of training children twice, several years apart. The double examination of the children would make it possible to accurately determine how the training of a given sport at different ages influences the shape of the spine’s curves. One-time examination of the same group of children only helps to characterize the size of the spine’s curvature by indication of small dependencies such as somatic conditions, training period and the type of exercises that affect the parameters of body posture. Therefore, the direction of future research should be to compare the same study group over several years. This will allow the unequivocal assessment of the impact of training the selected sports has on the shape of the spine’s curves. The limitation of the study may also be the subjective determination of bone points on the child’s body by the researcher. On the other hand, the examined children had a low BMI and the presence of only one researcher and one study can be considered a strength of the study. In order to confirm the effectiveness and precision of the study, the repeatability of measurements was assessed using the ICC (Intraclass Correlation Coefficient). Additionally, the limitation of the research was the combination of children training biathlon and taekwondo into one research group. The direction of future research should be to collect a sufficient research group that would allow us to independently assess the impact of the above sport disciplines on body posture.

## 5. Conclusions

Most of the respondents had the correct body posture in the sagittal plane regardless of the type of sport practiced.

Our results did not allow us to unequivocally state whether practicing various sports and having different training loads resulting from these sports had a negative or positive effect on the size of the anterior–posterior curvatures of the spine.

## Figures and Tables

**Figure 1 ijerph-20-01826-f001:**
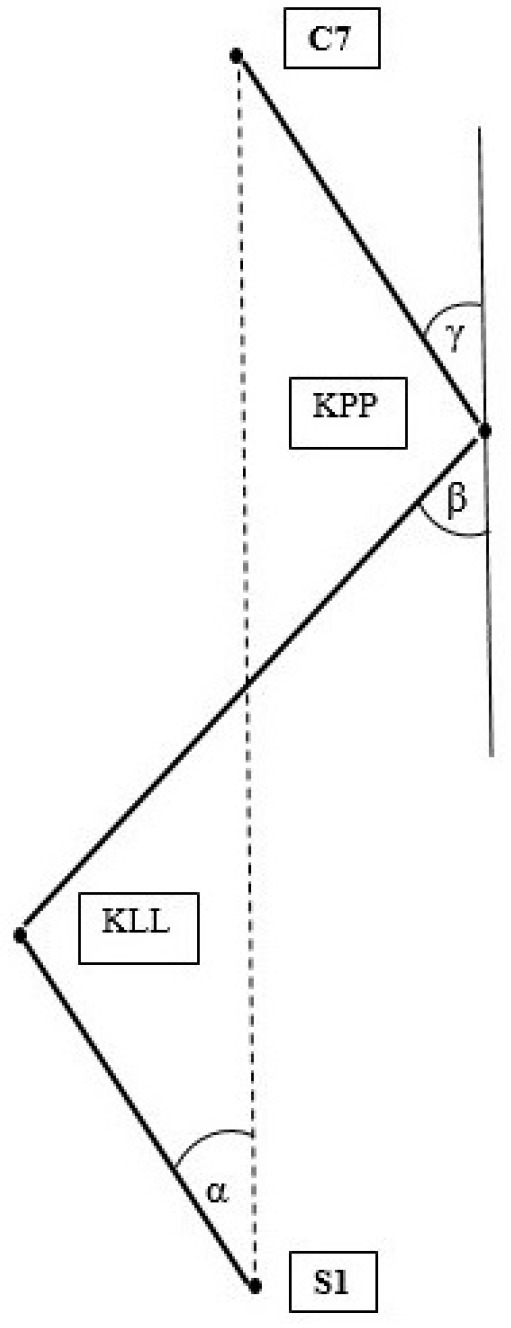
Summary of parameters for the assessment of anterior–posterior curvatures of the spine. Source: Own material.

**Figure 2 ijerph-20-01826-f002:**
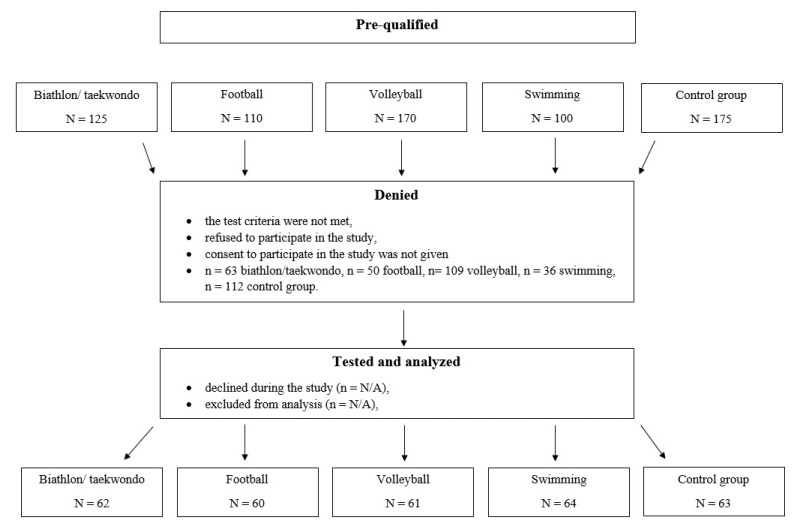
A diagram showing the participation of the respondents in the subsequent stages of the project.

**Figure 3 ijerph-20-01826-f003:**
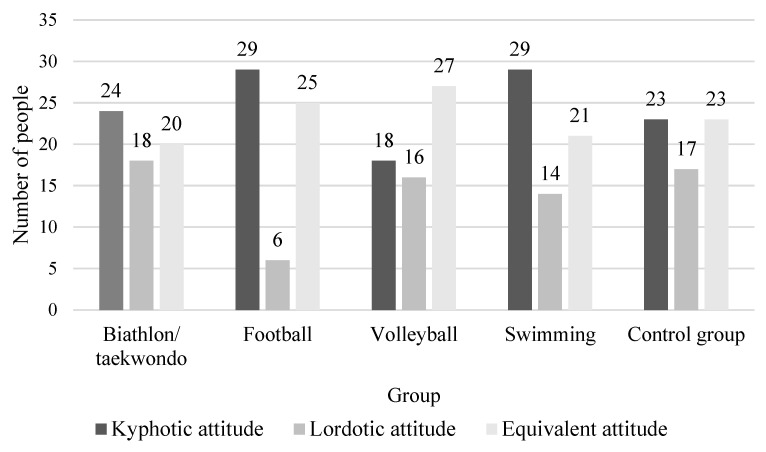
Types of body posture in the studied groups.

**Figure 4 ijerph-20-01826-f004:**
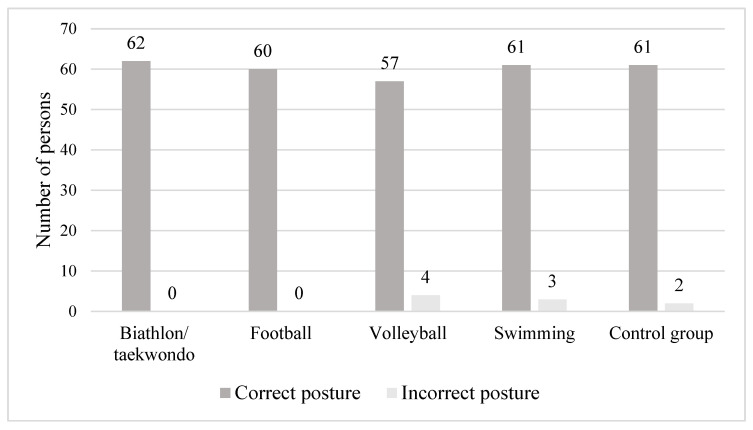
Characteristics of correct and incorrect postures based on the curvature of the spine in the studied groups.

**Table 1 ijerph-20-01826-t001:** Biometric data of the examined children.

		Group					
Parameter	Sex	Biathlon/Taekwondo	Football	Volleyball	Swimming	Control	*p*
*n* [number of persons]	F	29	0	44	32	28	-
	M	33	60	17	32	35	
	Total	62	60	61	64	63	
Age [years]	F	10.7 ± 1.3	-	11.9 ± 1.0	10.3 ± 0.7	11.2 ± 1.1	*p* < 0.0001
	M	11.1 ± 1.3	11.0 ± 1.0	11.4 ± 1.2	10.4 ± 0.9	11.3 ± 1.2	
	Total	10.9 ± 1.3	11.0 ± 1.0	11.8 ± 1.1	10.3 ± 0.8	11.3 ± 1.1	
Body height [cm]	F	145.4 ± 7.4	-	157.9 ± 11.2	143.7 ± 8.0	150.4 ± 9.3	*p* < 0.0001
	M	148.5 ± 10.2	147.0 ± 8.6	150.2 ± 7.3	147.8 ± 8.0	151.7 ± 8.7	
	Total	147.0 ± 9.1	147.0 ± 8.6	155.8 ± 10.8	145.7 ± 8.2	151.1 ± 8.9	
Body weight [kg]	F	37.1 ± 6.3	-	45.8 ± 11.5	35.2 ± 7.7	42.9 ± 8.1	*p* < 0.0001
	M	39.6 ± 8.2	37.0 ± 8.3	37.1 ± 5.9	38.2 ± 6.7	43.9 ± 10.8	
	Total	38.4 ± 7.4	37.0 ± 8.3	43.4 ± 10.9	36.7 ± 7.3	43.4 ± 9.6	
BMI [kg/m^2^]	F	17.5 ± 1.9	-	18.1 ± 2.6	16.9 ± 2.2	18.8 ± 2.3	0.0013
	M	17.8 ± 2.2	16.9 ± 2.5	16.4 ± 1.4	17.4 ± 2.3	18.9 ± 3.6	

Statistically significant differences in selected parameters between the groups were observed.

**Table 2 ijerph-20-01826-t002:** Characteristics of the Alpha, Beta and Gamma parameters.

Group	$\mathrm{Descriptive}\;\mathrm{Statistics}\;(\bar{x}\;\pm \mathrm{SD})$				
	Alpha Angle	Beta Angle	Gamma Angle	KPP	KLL
Biathlon/taekwondo	11.18 ± 4.54	4.22 ± 2.60	12.83 ± 5.11	162.95 ± 5.57	164.60 ± 4.59
Football	11.50 ± 4.39	3.18 ± 2.27	15.03 ± 3.58	161.79 ± 3.66	165.32 ± 5.22
Volleyball	12.17 ± 3.97	4.45 ± 3.38	12.05 ± 3.57	163.50 ± 4.79	163.38 ± 4.61
Swimming	12.30 ± 4.76	3.68 ± 2.33	13.58 ± 3.69	162.74 ± 4.01	164.02 ± 5.72
Control	13.44 ± 4.53	3.17 ± 2.48	14.37 ± 2.96	162.46 ± 3.76	163.39 ± 4.68
*p*	0.0496	0.0165	0.0002	0.2977	0.1510
Power	60%	60%	90%	43%	38%

## Data Availability

Raw data of this article are available upon request to the corresponding author.

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
