# Peer review of "The Sizes of Spine Curvatures of Children That Practice Selected Sports"

_ijerph, 2023, doi:10.3390/ijerph20031826_

Round 1
Reviewer 1 Report
Congratulations to the authors on this interesting and equally relevant study. In fact, it indicates which sports may lead to e.g. kyphotic posture.
Of course, it'd be even more interesting to have a follow-up study to further determine the effect. Nevertheless, the manuscript needs revision in regard to the use of English.
The aim was to determine the spinal curves in children practicing different sports, which in turn gives hints which sports may predispose to kyphotic posture and potential resulting conditions. Therefore the approach is quite interesting and moreover, their screening method is not radiography-based which yields obvious health benefits. Given the rising number of children exercising in a competitive manner as semi-professionals, this topic becomes more and more important. Furthermore, this research manuscript could serve as a base for future follow-up studies to examine the influence of different sports on spinal curvature. This study is purely descriptive – this may be stated by the authors – otherwise “controls” are OK. Not every study needs a “positive” outcome, it’s equally important to publish “negative” results in my humble opinion.
figure 3, kyphotic not kiphotic
page 7 "The results of own research allowed to determine the average values of the Alpha, Beta and Gamma angles, i.e. the angles determining the curvature of the spine, and it was found that the values of the Alpha and Gamma angles were within the normative ranges given for the same age" - rephrase! "In the own research, children have been training volleyball" try "our results suggest..."
Overall I'd recommend having this manuscript revised by an English native speaker to improve style and comprehensiveness.
Author Response
Thank you very much for all your valuable comments. All your suggestions - which we really appreciate - will certainly improve the quality of our article. Please find below the list of changes that we have made in our publication.
- Sentence reworded (250-252)
„The results of own research allowed to determine the average values of the Alpha, Beta and Gamma angles, i.e. the angles determining the curvature of the spine, and it was found that the values of the Alpha and Gamma angles were within the normative ranges given for the same age.” has been modified in the following way:
“The results of own research allowed to determine the average values of the angles Alpha, Beta and Gamma, i.e. the angles that define the curvature of the spine. The values of the angles fall into the normative ranges given for a specific age.”
- Conclusion reworded (324-326)
“The results of the author’s own research did not make it possible to unequivocally state whether practicing various sports and having different training load had a negative or positive effect on the size of the anterior-posterior curvatures of the spine.” has been modified in the following way:
“Our results do not allow to unequivocally state whether practicing various sports and having different training load resulting from these sports had a negative or positive effect on the size of the anterior-posterior curvatures of the spine.”
- Sentence reworded (283-287)
“Another criterion that could explain the differences between the results of the author's own research and that of Grabara [18] was training length. In the own research, children have been training volleyball for about 2 years, and in Grabara's work, the criterion for inclusion in the study is the training period lasting from 2 to 6 years. However, it cannot be clearly stated how the differences between Grabara's results and the results of own research were influenced by a different number of volleyball training sessions.” has been modified in the following way:
“Another criterion that could explain the differences between our results and that of Grabara [18] was the training period. Our research refers to children that have been training volleyball for about 2 years, while in Grabara's work, the criterion for inclusion in the study was the training period varying from 2 to 6 years. However, it cannot be clearly stated in what way the differences between Grabara's results and our results were influenced by a different number of volleyball training sessions.”
- The word “kyphosis” in the figure 3 has been corrected.
- Additionally, the whole publication has been verified by the translator.
Natalia Twarowska-Grybalow

Reviewer 2 Report
Comments and Suggestions for Authors
The manuscript entitled “The Sizes of Spine Curvatures of Children that Practice Selected Sports” is an original article that evaluates the shape of the spine curves in the cervical, thoracic and lumbar sections of the children that practice selected sports.
In my opinion, the aim of this manuscript is interesting because the authors tried to evaluate the influence of competitive sports on body posture in a phase of bone growth when the spine and the skeleton is still modifiable by muscle loads.
The article needs major revisions in order to be suitable for publication in the journal “IJRPH”.
Major Points:
First of all, the manuscript needs to have the lines numbered! I attached a file pdf where the article has numbered lines and my review will refer to this text.
Introduction
49: I would suggest authors to include a brief description of the gold standard methods used in literature to study the spine and body posture on the sagittal plane, introducing the reason of the choice of your instrumental method
Methods
57: why did you mix biathlon and taekwondo players? These two sports are different regarding the training program and the sporting gesture!
Have all other diseases caused muscle-skeletal deformities or structural postural deviations been excluded?
71-72: indicate the method used for evaluating the asymmetry of the length and scoliosis
Table 1: table caption/note are missing (among which parameters the p-value was calculated?
Instead of W and M, I would write Female (F) and Males (M)
Results
130-131: indicate the reference and values about correct and incorrect posture.
Discussions
240-245: About Limitation, you should insert the limitation relating to the instrumental method.
Minor points:
Abstract:
11-12: of the body posture with the Moiré method allowed for the assessment of the angles determining.
13: indicate the studied groups (football, swimming, taekwondo, volleyball).
Introduction
39-41: When mentioning " the risk of posture asymmetry.." I would suggest to add the following reference (doi: 10.23736/S0022-4707.21.12040-7. Epub 2021 Jun 17. PMID: 34137570. Long-term effects of asymmetrical posture in boxing assessed by baropodometry. J Sports Med Phys Fitness. 2022 Mar;62(3):350-355.).
Discussions
179: you can insert the aforementioned article about asymmetric sport [doi: 10.23736/S0022-4707.21.12040-7]

Author Response
Thank you very much for all your valuable comments. All your suggestions - which we really appreciate - will certainly improve the quality of our article. Please find below the list of changes that we have made in our publication.
- Added a brief description of the gold standard methods used in literature to study the spine and body posture on the sagittal plane, introducing the reason of the choice of your instrumental method (57-65)
Currently, a commonly used imaging test, evaluating deviations within the spine, which directly affects the entire body posture, is a radiographic examination. It is the optimal method for the analysis of the skeletal system [17,18]. X-ray examination is accurate and reliable, but it is associated with radiation and negative health effects resulting with frequent exposure. The projection method with the Moire system is a method of assessing body posture, supplementing the X-ray examination of the spine. The method uses an optical phenomenon, which is the process of refraction of light beams, which enables the creation of a three-dimensional image from a two-dimensional image [17]. The credibility of the research method using the Moire system has been confirmed by numerous publications in recent years [6,17,18,19].
- Explaining why biathlon and taekwondo groups merged despite differences in training program and the sporting gesture (76-79)
“biathlon/ taekwondo (62 persons attending the same class in a sport-profile primary school. All persons participated in these physical education classes and had the same general development training programme. Only the training programme of the leading sport discipline differed, and the number of training units depended on the trainees’ age)”
- Rewording the exclusion criteria for the study (96-103)
- severe orthopedic injuries (fracture of the lower limb, inability to fully load the limb),
- asymmetry of the length of the lower limbs above 2 cms (assessment by physical examination),
- structural scoliosis (assessment by physical examination),
- the presence of other diseases causing musculoskeletal deformities or structural postural defects
- failure to obtain the written consent of the legal guardian to conduct the examination.
- Added information and corrected the gender indication in the table (87-88)
“Statistically significant differences in selected parameters between the groups were observed.”
- The paragraph on body posture types and correct and incorrect postures, determined using the Wolański method, modified by Zeyland-Malawka, has been supplemented with additional information (161-163, 176-180)
On the basis of the mean values of angles and the size of curvatures in the spine, the types of body posture and the presence of correct or incorrect body posture in the study groups were determined by using the Wolański metho, modified by Zeyland-Malawka.
The kyphotic, lordotic and equivalent posture was also analyzed on the basis of the Wolański method, modified by Zeyland-Malawka and the attitudes were divided into correct and defective ones on the basis of the exact types of posture (I, II, III). The correct body posture is determined by type I of the equivalent posture, type II of the kyphotic posture and type I of the lordotic posture.
6. Adding a sentence to the Discussion on Research Limitations (313-318)The limitation of the study may also be the subjective determination by the researcher of bone points on the child's body. On the other hand, the examined children had a low BMI and the presence of only one researcher making one study can be considered a strength of the study. In order to confirm the effectiveness and precision of the study, the repeatability of measurements was assessed by using the ICC (Intraclass Correlation Coefficient). The coefficient ranged from 0.85 to 0.95.
7. Rephrasing of the sentence (10-12) „The study of the body posture that was made by using the Moiré method allowed for the assessment of the size of the angles determining the size of the spine's curvature” has been modified in the following way:”The study of the body posture with the Moiré method allowed for the assessment of the angles that determine the size of the spine's curvature”.
8. Rephrasing of the sentence (12-14)The analysis of differences between the groups included in the study (football, swimming, biathlon/ taekwondo, volleyball) was carried out on the basis of one-dimensional models taking into account the distributions of individual parameters.
9. Added the following reference: DE Blasiis, P.; Fullin, A.; Caravaggi, P.; et al. Long-term effects of asymmetrical posture in boxing assessed by baropodometry. J Sports Med Phys Fitness 2022, 62(3), pp. 350-355 Natalia Twarowska-Grybalow

Round 2
Reviewer 2 Report
Comments and Suggestions for Authors
The manuscript entitled “The Sizes of Spine Curvatures of Children that Practice Selected Sports” is suitable to publish in journal “IJRPH” with minor revisions.
Minor Points:
In Introduction, I would suggest modifying the previously integrated text according to the following revision and adding the mentioned references:
Currently, a commonly used imaging test for evaluating deviations within the spine, which directly affects the whole-body posture, is a radiographic examination. It is the gold standard method for the analysis of the skeletal system [17,18]. X-ray examination is accurate and reliable, but it is associated with radiation and negative health effects resulting with frequent exposure. The projection method with the Moire system is a method of assessing body posture, whose reliability has been confirmed by numerous publications in recent years [6,17,18,19]. This instrumental analysis uses an optical phenomenon, that is the process of refraction of light beams, which enables the creation of a three-dimensional image from a two-dimensional image [17]. Other instrumental investigations, such as 3D-motion analysis, quantitatively and three-dimensionally evaluate sagittal posture and spine in upright standing [doi: 10.3233/JND-210663. “Quantitative Evaluation of Upright Posture by x-Ray and 3D Stereophotogrammetry with a New Marker Set Protocol in Late Onset Pompe Disease.” J Neuromuscul Dis. 2021;8(6):979-988] and during walking [doi: 10.3390/jfmk7030057. Kinematic Evaluation of the Sagittal Posture during Walking in Healthy Subjects by 3D Motion Analysis Using DB-Total Protocol” J Funct Morphol Kinesiol. 2022 Aug 11;7(3):57 ]. Digital photography Moire method and 3D-Motion analysis don’t use radiation, show a good accuracy, but the former takes shorter test execution times, while the latter one also allows a dynamic study. However, they are supplementary examinations to the radiological examination.
Finally, the reasons reported about the similarity of the two sports (biathlon and taekwondo) are not convincing. I would include this concept in the limitations of the study
Author Response
Thank you very much for further comments. All are included in the text
1. Rephrasing paragraph about Moire system as suggested2. Added the following references:
De Blasiis, P.; Fullin, A.; Sansone M.; et al. Quantitative Evaluation of Upright Posture by x-Ray and 3D Stereophotogrammetry with a New Marker Set Protocol in Late Onset Pompe Disease. J Neuromuscul Dis 2022, 8(6), pp. 979-988.
De Blasiis, P.; Fullin, A.; Sansone M.; et al. Kinematic Evaluation of the Sagittal Posture during Walking in Healthy Subjects by 3D Motion Analysis Using DB-Total Protocol. J Funct Morphol Kinesiol 2022, 7(3), p:57
3. Adding a sentence to the Discussion on Research LimitationsAdditionally, the limitation of the research is the combination of children training biathlon and taekwondo into one research group. The direction of future research should be to collect a sufficient research group that would allow to assess independently the impact of the above sports disciplines on body posture.
Natalia Twarowska-Grybalow